# Different Oncologic Outcomes According to Margin Status (High-Grade Dysplasia vs. Carcinoma) in Patients Who Underwent Hilar Resection for Mid-Bile Duct Cancer

**DOI:** 10.3390/cancers15215166

**Published:** 2023-10-26

**Authors:** Hani Jassim Alramadhan, Soo-Yeun Lim, Hye-Jeong Jeong, Hyun-Jeong Jeon, Hochang Chae, So-Jeong Yoon, Sang-Hyun Shin, In-Woong Han, Jin-Seok Heo, Hongbeom Kim

**Affiliations:** 1Department of Surgery, King Fahad Hospital, Hofuf 36411, Saudi Arabia; dr.hani-jar@hotmail.com; 2Division of Hepatobiliary-Pancreatic Surgery, Department of Surgery, Samsung Medical Center, School of Medicine, Sungkyunkwan University, Seoul 06351, Republic of Korea; sooyeun.lim.x@gmail.com (S.-Y.L.); hyejeong.jeong@samsung.com (H.-J.J.); mlhjdream@naver.com (H.-J.J.); hochang.chae@samsung.com (H.C.); sojeong.yoon@samsung.com (S.-J.Y.); surgeonssh@gmail.com (S.-H.S.); cardioman76@gmail.com (I.-W.H.); jinseok.heo@samsung.com (J.-S.H.)

**Keywords:** bile duct cancer, cholangiocarcinoma, high-grade dysplasia

## Abstract

**Simple Summary:**

For hilar cholangiocarcinoma (HCCA), margin positivity after hilar resection for bile duct cancer is commonly observed due to its longitudinal spread along the subepithelial plane; however, we cannot draw conclusions regarding the prognostic effects of margins with high-grade dysplasia (HGD) or carcinoma. This study aimed to investigate the significance of positive bile duct margins and indications for extended resection in patients with middle bile duct cancer and to analyze the incidence of the R1 HGD margin and its clinical significance. We found that overall survival and disease-free survival in the R1 HGD–CIS margin were comparable with the R0 margin and significantly better than the R1 carcinoma. Extended resection should be considered in patients with R1 carcinoma-positive margins; however, extended resection in cases of R1 HGD-positive margins may not be necessary.

**Abstract:**

Margin positivity after hilar resection (HR) for bile duct cancer is commonly observed due to its longitudinal spread along the subepithelial plane; nevertheless, we cannot draw conclusions regarding the prognostic effects of margins with high-grade dysplasia (HGD) or carcinoma. We aimed to investigate the oncologic effect according to the margin status after HR, particularly between the R1 HGD and the R1 carcinoma. From 2008 to 2017, 149 patients diagnosed with mid-bile duct cancer in Samsung Medical Center, South Korea, were divided according to margin status after HR and retrospectively analyzed. Recurrence patterns were also analyzed between the groups. There were 126 patients with R0 margins, nine with R1 HGD, and 14 with R1 carcinoma. The mean age of the patients was 68.3 (±8.1); most patients were male. The mean age was higher in R1 carcinoma patients than in R1 HGD and R0 patients (*p* = 0.014). The R1 HGD and R1 carcinoma groups had more patients with a higher T-stage than R0 (*p* = 0.079). In univariate analysis, the prognostic factors affecting overall survival were age, T- and N-stage, CA19-9, and margin status. The survival rate of R0 was comparable to that of R1 HGD, but the survival rate of R0 was significantly better compared to R1 carcinoma (R0 vs. R1 HGD, *p* = 0.215, R0 vs. R1 carcinoma, *p* = 0.042, respectively). The recurrence pattern between the margin groups did not differ significantly (*p* = 0.604). Extended surgery should be considered for R1 carcinoma; however, in R1 HGD, extended operation may not be necessary, as it may achieve oncologic outcomes similar to R0 margins with HR.

## 1. Introduction

Cholangiocarcinoma (CCA) is a rare tumor that can occur along the biliary tree. It accounts for 3% of gastrointestinal malignancies [1]. It can be classified based on its anatomic location as an intrahepatic or extrahepatic CCA, which includes both perihilar and distal CCA. Hilar cholangiocarcinoma (HCCA) was first described by Klatskin in 1965 and is the most common form of CCA, accounting for 50% of all cases [2]. A complete resection provides patients with the highest chance of a cure [3]. CCA is characterized by longitudinal spread along the bile ducts in the subepithelial plane that can extend up to 2 cm proximally and 1 cm distally [4].

A positive microscopic resection margin is frequently observed and can be classified into invasive or noninvasive components (carcinoma in situ) [5,6,7]. Resection margin status is one of the strongest prognostic factors affecting survival. Patients with positive margins had significantly reduced survival rates compared to patients with negative margins [5]. It is a general practice that residual invasive cancer at a surgical margin requires additional hilar resection (HR), but the need for additional resection in the case of an R1 high-grade dysplasia (HGD) margin and its clinical significance remain unclear. Liver and pancreatic resection are considered to be the first-line approaches for patients with Bismuth type III–IV or distal CCA, respectively. However, the optimal surgical procedure for patients with Bismuth I and II remains controversial and debatable.

A number of studies have advocated concomitant hepatic resection for type I and II HCCA as a means to increase R0 rates and decrease the incidence of local recurrence, thus providing more favorable outcomes than HR alone [8]. However, major liver resection carries a high morbidity and surgery-related death rate (mortality, 10%) [9]. On the other hand, several studies have shown no significant difference between HR and hepatectomy in terms of R0 resection margin rate and survival [10,11].

This study aimed to investigate the significance of positive bile duct margins and indications for extended resection in patients with mid-bile duct cancer and to analyze the incidence of the R1 HGD margin and its clinical significance.

## 2. Patients and Methods

### 2.1. Patients

A prospectively maintained database of all patients with mid-bile duct cancer who underwent surgical resection with curative intent between 2008 and 2017 was included. Patients were excluded if the surgery was aborted because of occult metastasis or locally advanced disease. Patients with macroscopically positive resection margins (R2) were also excluded (Figure 1).

### 2.2. Surgical Procedure

Preoperatively, the location of the tumor and the extent of the tumor along the biliary tract were evaluated using imaging studies, including enhanced computed tomography (CT), ultrasonography, magnetic resonance imaging (MRI), and endoscopic retrograde cholangiopancreatography (ERCP).

Surgical procedures were decided by each attending surgeon based on tumor location and extension, the margin status of the frozen section, and patient operative risk. After laparotomy and the exclusion of distant metastasis, a regional lymphadenectomy was performed on the right side of the celiac artery, and all tissues in the hepatoduodenal ligament were removed (skeletonization of the hepatoduodenal ligament), except for the portal vein and the hepatic artery. In patients with Bismuth type I–II HCCA and supra-pancreatic distal CCA, limited HR was performed. Intraoperative bile duct frozen sectioning of the proximal (hepatic)-side and/or distal (duodenal)-side ductal margins was performed in all patients. When the distal-side ductal margin was positive, additional resection of the intrapancreatic bile duct, or PD, was performed. When the proximal-side ductal margin was positive, additional resection of the hepatic duct, hepatectomy, or pancreaticoduodenectomy was performed.

### 2.3. Diagnosis and Definition of Surgical Margins

Resected specimens were submitted to the Department of Pathology for histological evaluation, upon which experienced hepatobiliary pathologists examined all the specimens. Based on the type of resection performed, the appropriate proximal and distal bile duct margins were identified, and a cross-section of each was examined. Microscopically-positive surgical margins were classified into two categories: ‘carcinoma’ and ‘HGD’. Radial margins were defined as surgical margins other than the ductal margins of the resected specimen.

### 2.4. Comparison of Clinicopathological Variables and Patient Follow-Up

Clinicopathological variables including age, sex, location of a positive margin, and histological grade and type were evaluated. Histological findings were described in accordance with the TNM staging system of the American Joint Committee on Cancer, seventh edition.

Patients were followed up regularly in outpatient clinics every three–six months, and information during follow-up was obtained for all patients. The sites of initial disease recurrence were determined using cross-sectional imaging studies, such as CT or MRI. They were classified as local disease recurrence (resection margin, bilioenteric anastomosis, porta hepatis, or regional lymph nodes) and systemic disease recurrence (intrahepatic, peritoneal, or extra-abdominal sites). Overall survival (OS) was analyzed from the date of surgical resection to the date of death from all causes. The causes of death were determined from medical records. The follow-up period was defined as the interval between the date of surgical resection and the last follow-up.

### 2.5. Adjuvant Treatment

At stage ≥2, margin-positive patients were referred to a medical oncologist for adjuvant treatment; however, the final decision regarding adjuvant treatment, regimen, and cycle was made on an individual basis by a medical oncologist.

### 2.6. Statistical Analysis

Mean and standard deviation were used for continuous variables, whereas categorical variables were expressed as numbers and proportions. Tumor markers are expressed as median values and interquartile ranges (IQRs). Categorical variables were compared using the Student’s *t*-test and the X2 test. Continuous variables were compared using the independent *t*-test or Mann–Whitney U-test. Kaplan-Meier analysis was used to analyze OS and disease-free survival (DFS). Differences in the survival curves were compared using log rank tests. The Cox proportional hazards model was used to identify the factors independently associated with postoperative survival. Any *p*-values of 0.05 or less were considered statistically significant. Data were analyzed using IBM SPSS Statistics for Windows, version 27 (IBM Corp., Armonk, NY, USA). This study was conducted with approval from the Institutional Review Board of the Samsung Medical Center (IRB No. 2022-11-098).

## 3. Results

### 3.1. Demographics and Margin Status

A total of 149 patients were included: 104 males (69%) and 45 females (31%), with a median age of 68.3 years. The demographic and clinicopathological data of the entire cohort are presented in Table 1. There were 126 patients in the R0 group, nine in the R1 HGD group (6%), and 14 in the R1 carcinoma group (9.3%). Information on margin analysis is presented in Appendix A. The presence of R1 HGD at the resection margin was significantly high with a large primary tumor size (*p* = 0.079). Adjuvant chemotherapy was prescribed to 12 patients in the R0 group (9.5%), none in the R1 HGD group, and two patients in the R1 carcinoma group (14.3%). Adjuvant radiotherapy was administered to 33 patients in the R0 group (22.1%), three patients in the R1 HGD group (33.3%), and five patients in the R1 carcinoma group.

### 3.2. Survival Analysis and Prognostic Factors for Survival

OS in the R0 group was significantly better than that in the R1 carcinoma group (five-year OS: 53.3% vs. 23.8%, respectively; *p* = 0.042); however, there was no significant difference in the five-year OS between the R0 and R1 HGD groups (five-year OS: 53.3% vs. 27.8%, respectively; *p* = 0.215) (Figure 2a); moreover, there was no significant difference in DFS between the R0 and R1 carcinoma groups (32.5 months (35.2%) vs. 21.1 months (9.8%), respectively; *p* = 0.230). The five-year DFS was also not significantly different between the R0 and R1 HGD groups (32.5 months (35.2%) vs. 26.1 months (14.8%), respectively; *p* = 0.230) (Figure 2b).

Multivariate Cox proportional hazards analysis revealed that preoperative CA19-9 level, T-stage, N-stage, and resection margin status were independent prognostic factors for OS (Table 2). The factors that influenced DFS were CA19-9 level, T stage, and N stage (Appendix A).

### 3.3. Recurrence Patterns

A recurrence was observed in 84 patients (56%) during follow-up. 45 patients had local recurrence, while 39 had systemic metastasis. Local and systemic recurrences were analyzed based on the margin status. Local recurrence was found in 39 out of 71 patients with R0 margins (54.9%), four out of seven patients with R1 HGD (57.1%), and two out of six patients with R1 carcinoma (33.3%). Systemic recurrences occurred in 32 (45.1%), three (42.9%), and four (66.7%) patients in the R0, R1 HGD, and R1 carcinoma groups, respectively. There was no significant relationship between ductal margin status and disease recurrence (*p* = 0.274). Lymph nodes were the main sites of local recurrence, and the liver was the main site of systemic metastasis in the three groups. Details regarding recurrence patterns are presented in Table 3.

### 3.4. Clinical Course of R1 HGD Patients

Seven out of the nine (77.8%) patients with R1 HGD margins were found to have disease recurrence during follow-up. Local recurrence was observed in four patients (57.1%), all in the lymph nodes. No patients in this group experienced local recurrence within the duct anastomosis site. Three patients (42.9%) were found to have systemic disease, out of which two had liver metastasis along with peritoneal seeding and one had liver metastasis with lymph node recurrence. Table 4 outlines the clinical course of the R1 HGD group.

## 4. Discussion

CCA is a rare, aggressive tumor. A complete resection with negative margins provides patients with the greatest chance of a cure [1,3]. Preoperative determination of tumor extension can be difficult due to the tendency of CCA to spread longitudinally along the bile duct in the subepithelial plane. Tumors may extend up to 2 cm proximally and 1 cm distally and can be classified as invasive or noninvasive components (carcinoma in situ) [5,6,7]. The reported incidence of carcinoma in situ (CIS) at the resection margin is 3–16% [12,13,14,15]. Several studies have reported that patients with residual invasive cancer at the resection margin have significantly worse outcomes than those with a negative resection margin (R0). However, several authors have reported no survival difference when the outcomes of patients with residual carcinoma in situ at the margin were compared with those of an R0 resection margin. However, the number of patients included in these studies was small [13,16,17,18,19,20,21]. Many of the previous studies included bile duct cancers from different locations along the biliary tree (intrahepatic, hilar, gallbladder, distal, etc.), which might have different biological behavior and require different surgical procedures that can alter clinical outcomes [5,15,22]. In the current study, we analyzed patients with mid-bile duct cancer who underwent a limited HR. We demonstrated similar findings: survival rates and oncological outcomes for the R1 HGD positive margin were comparable with R0 and significantly better than those of R1 carcinoma.

Some authors have reported an increase in long-term local recurrence in patients with R1 HGD-positive margins compared to R0, suggesting that HGD is a slow-growing lesion [12,13,15,17,18]. In contrast, some studies have reported a significant increase in local recurrence when patients were stratified into early cancer stages (T1–T2, N0, M0). This suggests that, in advanced disease (N1–2), the effect of CIS on prognosis is masked by more powerful factors. It can be concluded that, in early disease resection, achieving a negative margin is recommended if R1 HGD–CIS is encountered at the resection margin, while, in advanced node-positive disease, resection to achieve an R0 margin may not be necessary [12,15,18,23].

Controversies remain regarding the benefits of adjuvant treatment in cases of extrahepatic CCA. Several studies have demonstrated the beneficial effects of adjuvant chemotherapy on prognosis [24,25,26]. Lee et al. reported similar survival rates between R0 resection patients without adjuvant treatment and R1 resection patients who received gemcitabine-based chemotherapy (*p* = 0.6193) [26]. Watson et al. reported similar results [27]. There was no difference in DFS and OS between R0 and R1 patients who received adjuvant chemotherapy. These findings indicate that the use of modern chemotherapy may offset the negative effect of the R1 margin and that additional resection to achieve an R0 margin is not an absolute requirement. Further prospective clinical trials are required to confirm our findings.

The intraoperative frozen sectioning of margins has been used by many centers to guide the intraoperative extent of resection. Intraoperative frozen section results may differ from the final permanent section or “true margin” in up to 9–25% of cases. In our study, we performed frozen section biopsies in 100% of our cases, and the rate of difference between final pathology and frozen biopsy was 2.7%, which is below the published data [15,16]. In this study period, due to positive frozen section margins during HR, 53 patients (12.6%) had a pancreatectomy and 50 patients (20.4%) had a major liver resection. Four patients out of the R1 group received an intraoperative frozen section result of “high-grade dysplasia”; however, all four changed to “carcinoma” in the final pathology. In our study, the conversion rate was 2.7%, which is below the previous studies. A factor that can contribute to this change is the inflammation caused by tumor infiltration or post-biliary drainage procedures [11,17,18]. The characteristic longitudinal submucosal spread may also contribute to this discrepancy.

Although HR with liver resection or pancreaticoduodenectomy is accepted for the management of Bismuth type III–VI and distal bile duct cancer, surgical management of mid-bile duct cancer (Bismuth type I–II) is still debated. Several studies have shown that limited HR is sufficient in well-selected patients and has oncological outcomes similar to those of liver resection [11,28,29,30]; however, several authors have reported better long-term outcomes in patients who underwent liver resection [30,31,32]. Zhang et al., in their multi-institutional study of 257 patients with type I–II CCA, demonstrated that the incidence of R0 margins was similar between the limited HR and liver resection groups [11]. DFS and OS rates were comparable between the groups. In contrast, the incidence of severe postoperative complications was significantly higher after a major liver resection. The majority of previous studies were conducted in single centers with small sample sizes. Several factors may have led to these conflicting results, including differences in surgical technique, the inclusion of various Bismuth–Corlette HCCA subtypes rather than just type I and II lesions, and variation in the evaluation of frozen section specimens during surgery [14,15].

Our study has several limitations. This was a retrospective study; as such, selection bias may exist. Further, the number of patients was too small. Although it is a very rare condition, the evidence of survival comparison is very weak based on nine and 14 patients. We will advance our next multi-center study with an enlarged sample size to overcome this limitation. The study period was long, and indications and regimens of adjuvant therapy may have changed over time. The present study has several strengths, however. This is one of the largest single-center series reporting the effect of the margin status of mid-bile duct cancer after limited HR. The same pathologist and only a limited number of surgeons were involved, which maintained consistency in the surgical treatment.

## 5. Conclusions

In conclusion, the OS and DFR in the R1 HGD–CIS margin were comparable with the R0 margin and significantly better than those in R1 carcinoma. Extended resection should be considered in patients with R1 carcinoma-positive margins; however, extended resection in cases with R1 HGD-positive margins may not be necessary. Limited HR of mid-bile duct cancer may be oncologically adequate if an R0 margin can be achieved.

## Figures and Tables

**Figure 1 cancers-15-05166-f001:**
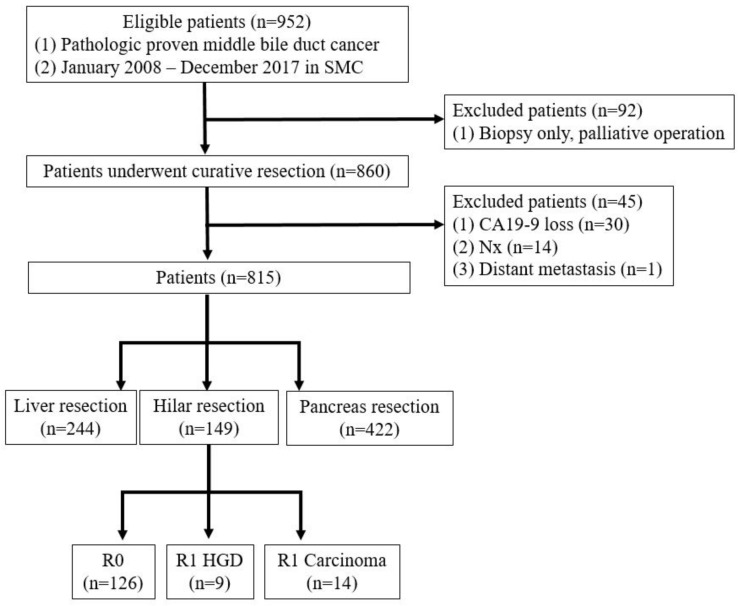
Patient selection.

**Figure 2 cancers-15-05166-f002:**
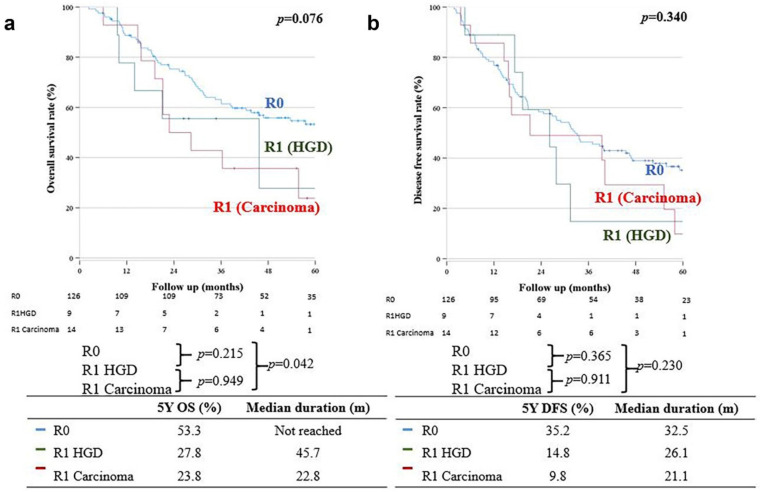
(**a**) Overall survival. (**b**) Disease-free survival.

**Table 1 cancers-15-05166-t001:** Demographic and clinicopathological characteristics (*n* = 149).

Variables N (%) or Mean (±sd)	Total (*n* = 149)	R0 (*n* = 126)	R1 HGD (*n* = 9)	R1 Carcinoma (*n* = 14)	*p*-Value
Sex (M:F)	104:45	90:36	4:5	10:4	0.252
Age	68.3 (8.1)	67.5 (8.2)	70.6 (5.0)	73.6 (7.3)	0.014
BMI	23.4 (3.3)	23.5 (3.4)	22.9 (2.9)	22.3 (1.7)	0.406
ASA					0.114
I	18 (12.1)	16 (12.7)	0 (0)	2 (14.3)	
II	113 (75.8)	98 (77.8)	7 (77.8)	8 (57.1)	
III/IV	18 (12.1)	12 (9.5)	2 (22.2)	4 (28.6)	
CA 19-9	29.2	29.2	63.9	27.7	0.911
Median (IQR)	(13.9–81.1)	(13.3–74.3)	(28.3–496.0)	(13.7–188.8)	
Postoperative hospital days	10.7 (7.9)	10.9 (8.4)	7.7 (1.0)	10.4 (3.8)	0.480
T-stage					0.079
T1	47 (31.5)	44 (34.9)	1 (11.1)	2 (14.3)	
T2a/T2b	91 (61.1)	75 (59.5)	7 (77.8)	9 (64.3)	
T3/T4	11 (7.4)	7 (5.6)	1 (11.1)	3 (21.4)	
N-stage					0.466
N0	107 (71.8)	91 (72.2)	6 (66.7)	10 (71.4)	
N1	39 (26.2)	33 (26.2)	2 (22.2)	4 (28.6)	
N2	3 (2.0)	2 (1.6)	1 (11.1)	0 (0)	
Complications					0.448
No	133 (89.3)	113 (89.7)	7 (77.8)	13 (92.9)	
Yes	16 (10.7)	13 (10.3)	2 (22.2)	1 (7.1)	
Adjuvant chemotherapy					0.702
No	135 (90.6)	114 (90.5)	9 (100)	12 (85.7)	
Yes	14 (9.4)	12 (9.5)	0 (0)	2 (14.3)	
Adjuvant radiotherapy					0.225
No	116 (77.9)	101 (80.2)	6 (66.7)	9 (64.3)	
Yes	33 (22.1)	25 (19.8)	3 (33.3)	5 (35.7)	

SD, standard deviation.

**Table 2 cancers-15-05166-t002:** Uni- and multivariate analysis identifying factors affecting overall survival (*n* = 149).

Variable	Patients (*n*)	5Y OS (%)	Univariate Analysis			Multivariate Analysis		
			HR	95% CI	*p*	HR	95% CI	*p*
Sex								
Male/Female	104/45	46.9/54.7	0.844	0.529–1.346	0.476			
Age								
≤65/>65	56/93	58.2/43.4	1.616	1.021–2.561	0.041	1.268	0.777–2.069	0.343
BMI								
≤25/>25	106/43	45.3/59.2	0.657	0.393–1.097	0.108			
ASA score					0.355			
I	18	50.3						
II	113	51.9	0.759	0.414–1.388	0.370			
III/IV	18	34.6	1.132	0.514–2.493	0.759			
Preop CA19-9								
≤35/>35	84/65	69.6/24.8	3.421	2.198–5.325	<0.001	2.618	1.610–4.257	<0.001
T-stage					<0.001			0.043
T1	47	82.2						
T2	91	35.5	3.810	2.094–6.933	<0.001	2.211	1.162–4.207	0.016
T3/4	11	34.1	4.126	1.704–9.993	0.002	2.547	0.975–6.652	0.056
N stage								
N (−)/N (+)	107/42	58.5/24.5	2.316	1.496–3.585	<0.001	1.587	0.989–2.547	0.055
Margin								
R0	126	53.3			0.082			0.149
R1 HGD	9	27.8	1.791	0.715–4.486	0.213	1.391	0.541–3.580	0.493
R1 carcinoma	14	23.8	1.916	1.008–3.640	0.047	1.932	0.982–3.799	0.056
Complications								
No/Yes	133/16	51.0/34.4	1.298	0.642–2.623	0.468			

**Table 3 cancers-15-05166-t003:** Summary of recurrence pattern (*n* = 149).

Variables	Total (*n* = 149)	R0 (*n* = 126)	R1 HGD (*n* = 9)	R1 Carcinoma (*n* = 14)	*p*-Value
Recur (−)	65 (43.6)	55 (43.7)	2 (22.2)	8 (57.1)	0.274
Recur (+)	84 (56.4)	71 (56.3)	7 (77.8)	6 (42.9)	0.274 0.604
Local	45 (53.6)	39 (54.9)	4 (57.1)	2 (33.3)	
Systemic	39 (46.4)	32 (45.1)	3 (42.9)	2 (33.3)	

**Table 4 cancers-15-05166-t004:** Clinical course of R1 HGD patients (*n* = 9).

No.	Sex	Age	CA19-9	T	N	TNM	R1	Adjuvant Treat	Recurrence	Recurrence Site	Status	DFS	OS
1	F	67	7.48	T2	N0	II	Proximal	RTx	No	(-)	Alive	63	63
2	M	74	63.93	T3	N0	IIIA	Proximal	RTx	Systemic	Liver and peritoneal seeding	Death	7	10
3	F	76	183.82	T2	N2	IVA	Distal	No	Local	LN	Death	31	46
4	M	81	622.72	T2	N0	II	Proximal	RTx	Local	LN	Alive	26	35
5	F	60	7.65	T2	N1	IIIC	Proximal	RTx	Systemic	Liver, Peritoneal seeding, and LN	Death	13	14
6	F	75	369.3	T2	N1	IIIC	Distal	RTx	Systemic	Liver and LN	Death	19	21
7	M	84	28.69	T1	N0	I	Distal	No	No	(-)	Alive	28	28
8	F	77	662.52	T2	N0	II	Distal	RTx	Local	LN	Death	5	10
9	M	79	34.81	T2	N0	II	Proximal	No	Local	LN	Alive	17	27

## Data Availability

The data set is not publicly available due to privacy.

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
