# Peer review of "Different Oncologic Outcomes According to Margin Status (High-Grade Dysplasia vs. Carcinoma) in Patients Who Underwent Hilar Resection for Mid-Bile Duct Cancer"

_cancers, 2023, doi:10.3390/cancers15215166_

Round 1
Reviewer 1 Report
Dear Authors, first, I want to apologise for my late review. Unfortunately, I encountered a couple of unexpected events. Your retrospective study about margin status after hilar resection for mid bile duct cancer is well written, with a clear introduction, aim, methods and results.
Unfortunately, I could not open Table S1, so I cannot find an answer to my main concern: I can see that your classification in the three groups was based on the definitive pathology, but how were the intraoperative frozen sections of all patients in the three groups? How many pancreatectomy or other major resections did you perform in the intraoperative positive margin patients? Would the rate of extensive surgery have impacted the survival rate? Please discuss.
In addition, the sample size is an issue. I understand this is a rare condition and that exclusion criteria limited the number of selected patients. Still, even if your is a referral centre, you should state in the limitations that results might be affected by the small sample size.
Apart from these concerns, the sentence in the abstract about overall survival should be corrected: the term "higher" sounds inappropriate:
"The survival rate of R0 was comparable to that of R1 HGD, but 37 it was significantly higher in R1 carcinoma (R0 vs. R1 HGD, p = 0.215, R0 vs. R1 carcinoma, p = 0.042, 38 respectively)."
I recommend accepting this manuscript after minor revisions.
Author Response
Dear reviewer.
Thanks for your precious review. We replied details to each of your questions.
- Unfortunately, I could not open Table S1, so I cannot find an answer to my main concern: I can see that your classification in the three groups was based on the definitive pathology, but how were the intraoperative frozen sections of all patients in the three groups? How many pancreatectomy or other major resections did you perform in the intraoperative positive margin patients? Would the rate of extensive surgery have impacted the survival rate? Please discuss.
Author’s reply : All of 149 patients did the intraoperative frozen biopsy. Only 4 patients out of 9 R1 patients had changed result of frozen biopsy from high grade dysplasia to carcinoma. 53 patients (12.6%) had pancreatectomy and 50 patients (20.4%) had major liver resection because of positive intraoperative margin. We discuss about the operation type (extended op vs Hilar resection) in our next study. In that study, we figure out that these operation type doesn’t have statistical differences in survival outcome (HR 0.929, 0.707-1.219, p=0.594). Also, we had additional review about the positive rate of intraoperative biopsy.
- In addition, the sample size is an issue. I understand this is a rare condition and that exclusion criteria limited the number of selected patients. Still, even if your is a referral centre, you should state in the limitations that results might be affected by the small sample size.
Author’s reply : We will add our limitation about the small sample size. In our next study, we will gather multiple center data to overcome this problem.
- Apart from these concerns, the sentence in the abstract about overall survival should be corrected: the term "higher" sounds inappropriate.: "The survival rate of R0 was comparable to that of R1 HGD, but 37 it was significantly higherin R1 carcinoma (R0 vs. R1 HGD, p = 0.215, R0 vs. R1 carcinoma, p = 0.042, 38 respectively)."
Author’s reply. : Thanks for the note. We change our expression in this sentence “higher” into “better”, to make our purpose clear.: “The survival rate of R0 was comparable to that of R1 HGD, but the survival rate of R0 had significantly better, compared to R1 carcinoma (R0 vs. R1 HGD, p = 0.215, R0 vs. R1 carcinoma, p = 0.042, 38 respectively).”
Thank you.

Reviewer 2 Report
The authors present an interesting single centre study on the impact on survival/ recurrence free survival of resection margin status in resected perihilar cholangiocarcinoma. The authors conclude that a resection margin with high-grade dysplasia results in the same survival outcome as R0-resection and better than R1-resection.
The conclusion is important and could potentially guide the clinician to make, for the patient, the proper decision not to proceed with a liver resection if hilar resection only is performed and frozen section shows high-grade dyplasia.
I have however some concerns about the generalizability of the results presented. My first concern is the selection of patients. As I understand, all patients presented in Figure 1, are patients with Bismuth I or II perihilar cholangiocarcinoma. I would like the authors to comment on the reasons for performing liver resection or not (that is hilar resection) in these patients. To what extent was this due to arterial overgrowth of tumour? Did this group contain patients with a positive frozen section after hilar resection and where a subsequent liver resection was made? The same goes for the group that underwent pancreaticoduodenectomy. Was the decision to proceed with this procedure based on frozen section analysis?
Another concern is the sensitivity/specificity of frozen section analysis, which the authors discuss in the 4thparagraph in the discussion. How well did the method of frozen section correlate with final pathological examination in the present study? Also, why was not additional resection attempted in patients with cancer on frozen section?
Although coming from a large centre, the number of patients on which the authors base their conclusions is very limited. Talking about survival differences when there are 9 and 14 patients in the different groups is very hazardous. Therefore, the description of the patients must be meticulous, so the reader can judge how big seems to be the (inevitable) selection bias. The authors conclude ‘We found that the overall survival and disease-free survival in the R1 HGD-CIS margin were comparable with the R0 margin’. Based on only 9 patients, I believe that this is a very bold statement.
Author Response
Dear reviewer.
Thanks for your precious review. I replied details to each of your questions.
- I have however some concerns about the generalizability of the results presented. My first concern is the selection of patients. As I understand, all patients presented in Figure 1, are patients with Bismuth I or II perihilar cholangiocarcinoma. I would like the authors to comment on the reasons for performing liver resection or not (that is hilar resection) in these patients. To what extent was this due to arterial overgrowth of tumour? Did this group contain patients with a positive frozen section after hilar resection and where a subsequent liver resection was made? The same goes for the group that underwent pancreaticoduodenectomy. Was the decision to proceed with this procedure based on frozen section analysis?
Author’s reply : Most of these extended operation is caused by intraoperative frozen biopsy. As the perihilar cholangiocarcinoma is hard to guess the range, converting into extended operation for margin negative is very common. 53 patients (12.6%) had pancreatectomy and 50 patients (20.4%) had major liver resection because of positive intraoperative margin. Most of its decision is based on frozen biopsy result, however, some of them is based on tumor gross-looking appearance as well.
- Another concern is the sensitivity/specificity of frozen section analysis, which the authors discuss in the 4thparagraph in the discussion. How well did the method of frozen section correlate with final pathological examination in the present study? Also, why was not additional resection attempted in patients with cancer on frozen section?
Author’s reply : All of 149 patients did the intraoperative frozen biopsy. Only 4 patients out of 9 R1 patients had changed result of frozen biopsy from high grade dysplasia to carcinoma. Based on patient’s comorbidity and condition, clinician decide the additional resection. For example, if the margin positive patient is old age, past history with angina, COPD, and liver cirrhosis, it is hard to attempt liver resection with this patients.
- Although coming from a large centre, the number of patients on which the authors base their conclusions is very limited. Talking about survival differences when there are 9 and 14 patients in the different groups is very hazardous. Therefore, the description of the patients must be meticulous, so the reader can judge how big seems to be the (inevitable) selection bias. The authors conclude ‘We found that the overall survival and disease-free survival in the R1 HGD-CIS margin were comparable with the R0 margin’. Based on only 9 patients, I believe that this is a very bold statement.
Author’s reply : We will add limitation about sample size. We will proceed our next study with multi-center study to enlarge the sample size to overcome this problem.
Thank you.
